

# Understanding Hydrologic Variability across Europe through Catchment Classification

Anna Kuentz[1], Berit Arheimer[1], Yeshewatesfa Hundecha[1], Thorsten Wagener[2, 3]

[1] Swedish Meteorological and Hydrological Institute, 601 76 Norrköping, Sweden
[2] Department of Civil Engineering, University of Bristol, BS8 1TR, Bristol, UK
[3] Cabot Institute, University of Bristol, UK

*Correspondence to*: Anna Kuentz (anna.kuentz@smhi.se)

**Abstract.** This study contributes to better understanding the physical controls on spatial patterns of pan-European flow signatures - taking advantage of large open datasets for catchment classification and comparative hydrology. We explored similarities in 16 flow signatures and 35 catchment descriptors across entire Europe. A database of catchment descriptors and selected flow signatures was compiled for 35 215 catchments and 1366 river gauges across Europe. Correlation analyses and stepwise regressions were used to identify the best explanatory variables for each signature resulting in a total of 480 regression models to predict signatures for similar catchments. Catchments were clustered and analyzed for similarities in flow signature values, physiography and for the combination of the two. From the statistical analysis, we found: (i) about 400 statistically significant correlations between flow signatures and physiography; (ii) a 15 to 33% (depending on the classification used) improvement in regression model skills using catchment classification vs the full domain; and (iii) 12 out of 16 flow signatures to be mainly controlled by climatic characteristics, while topography was the main control for flashiness of flow and low flow magnitude, and geology for the flashiness of flow.

Classifying catchments based on flow signatures or on physiographic characteristics led to very different spatial patterns, but a classification and regression tree (CART) allowed us to predict flow signatures-based classes according to catchment physiographic characteristics with an average percentage of 60% of correctly classified catchments in each class. As a result, we show that Europe can be divided into ten classes with both similar flow signatures and physiography. We noted the importance of separating energy-limited catchments from moisture-limited catchment to understand catchment behavior. For improved understanding, we interpreted characteristics in hydrographs, flow signatures, physiography and geographical location to define dominant flow-generating processes. We found that rainfall response, snow-melt, evapotranspiration, damping, storage capacity, and human alterations could explain the hydrologic variability across Europe. Finally, we discuss the relevance of these empirical results for predictions in ungauged basins across Europe and for dynamic modelling at the continental scale.



## 1 Introduction

Hydrological systems exhibit a tremendous variability in their physical properties and in the hydrological variables we observe such as streamflow and soil moisture patterns (Bloeschl et al., 2013). At the catchment scale, we assume (or at least hope) that the aggregated response behavior, e.g. the hydrograph, is related to average or dominating characteristics and that smaller scale differences are less relevant. This assumption is the basis for statistical hydrology where it allows us to regionalize certain flow characteristics related to floods or low flows. We generally make the same assumption in the search for a catchment classification framework where we want to group catchments that somehow exhibit similar hydrologic behavior (McDonnell and Woods, 2004). So far we have not yet found a widely accepted classification system though it is generally agreed upon that even the search for such an organizing principle is an important undertaking for hydrology (Wagener et al., 2007).

A range of approaches have been taken to organize the catchments we find across our landscape. Approaches include the use of physical and climatic characteristics (e.g. Winter 2001; Brown et al., 2013; Buttle, 2006; Leibowitz et al., 2016), or the use of hydrologic signatures (e.g. Ley et al., 2011, Olden et al., 2012; Sawicz et al., 2011; Singh et al., 2016), or by also including water quality (Arheimer et al., 1996; Arheimer and Lidén, 2000). The advantage of the first approach is that physical characteristics such as topography and land cover are now available for any location on earth, while the second approach groups catchments directly by the characteristic we mainly care about, i.e. their hydrologic behavior (see discussion in Wagener et al., 2007). The disadvantages are that the first framework does not ensure that physically/climatically similar catchments also behave similarly, while the second is not directly applicable to ungauged catchments. Ultimately, we believe that a catchment classification framework has to achieve both to be useful, i.e. it has to be applicable to any catchment and provide insight into its expected hydrological behavior.

Here we assume that flow signatures are a relevant way towards quantifying hydrological behavior and therefore form a sensible basis for a classification framework. They condense hydrologic information that is derived from streamflow observations (alone or in combination with other variables) (Sivapalan, 2005). The choice of the specific signatures used for classification can be guided by: (i) the attempt to describe basic hydrological behavior (e.g. Ley et al., 2011, Sawicz et al., 2011; Trancoso et al., 2016); (ii) the need to relate to societally relevant issues such as floods and droughts (Wagener et al., 2008); (iii) the objective to characterize ecologically relevant characteristics of the catchment response (e.g. Olden et al., 2012); or (iv) in relation to subsequent hydrologic modeling (Euser et al., 2012; Hrachowitz et al., 2014; Donnelly et al., 2016). Studying differences and similarities in flow signatures as well as in catchment characteristics will improve our understanding of hydrological processes under current and under potential future conditions (Sawicz et al., 2014; Berghuijs et al., 2014; Pechlivanidis and Arheimer, 2015; Rice et al., 2015). Linking catchment descriptors (physical and climatic) and hydrological response signatures enables the inclusion of ungauged basins and provides the potential for assessing environmental change impacts across large domains.





Despite the significant world-wide research performed during many decades to both understand and predict hydrologic variability using physiography, work has largely addressed small or medium-sized and pristine catchments when delineating regions of similar flow controls (e.g. Yaeger et al., 2012; Ye et al. 2012, Patil and Stieglitz, 2012). Often different studies have resulted in conflicting relationships between some catchment responses and some of their physiographic controls, as a result of catchment size and geographical location. For instance, some studies have found out that forest cover reduces catchment streamflow (e.g. Hundecha and Bárdossy, 2004; Brown et al., 2005; Buytaert et al., 2007), while an increase in streamflow has been found in some others (e.g. Bruijnzeel, 2004). It would, therefore, be worthwhile to identify the physiographic controls of catchment responses and their relationships using a consistent approach across a larger geographic domain, which is subdivided into catchments of different spatial scales. A large sample of observed data from different physiographical and hydrological conditions, enable comparative analysis of dominant drivers for flow generation (Falkenmark and Chapman, 1989). No study so far, to our knowledge, has applied the results from comparative hydrology at the continental scale, also including large rivers with human alteration and ungauged basins.

This study aims at exploring and understanding the physical controls on spatial patterns of pan-European flow signatures by taking advantage of large open datasets. Better understanding would enhance our ability to predict hydrological variables in ungauged catchments for more efficient water management. We explore the relationships between catchment descriptors and flow signatures by analyzing 35 215 catchments which cover a wide range of pan-European physiographic and anthropogenic characteristics. A database of catchment descriptors and selected flow signatures is estimated for all catchments and 1366 flow gauges across Europe, providing material for a first level of analyses of statistical and spatial distribution. Correlation analyses are subsequently used to identify the best explanatory variables for each signature and to build regression models for predictions in ungauged basins. Catchments are clustered and analyzed for similarities in flow signatures, physiography and combination of the two, to further improve the predictability and to detect similarities in flow generating processes across the large domain.

The ultimate aim of our study is to better understand hydrological patterns across the European continent guided by the following science questions:

1. To what extent can physiography explain similarities in flow signatures across Europe?
2. What spatial pattern can be derived from combining similarity in flow signatures and physiography across the European continent?
3. Which flow generating processes can be attributed to regions with similar flow signatures?

## 2 Data and Methods

### 2.1. Database of catchment descriptors and flow signatures

A database of physiographical characteristics (climate, physical and human alteration) was compiled for 35,215 European catchments with a median size of 214 km$^2$ (Fig. 1). The geographical domain (8.8 million km$^2$) was delineated according to




plat-tectonic boarders combined with catchment boarders of rivers all the way down to the European coast and to the Ural Mountains in the East.

For each catchment, 48 physiographical descriptors were assigned using upstream topography, climate, soil types, land use as well as geology from open data sources (Table 1). Descriptors were estimated as spatial means of the upstream area and
5  assigned to each catchment outlet.

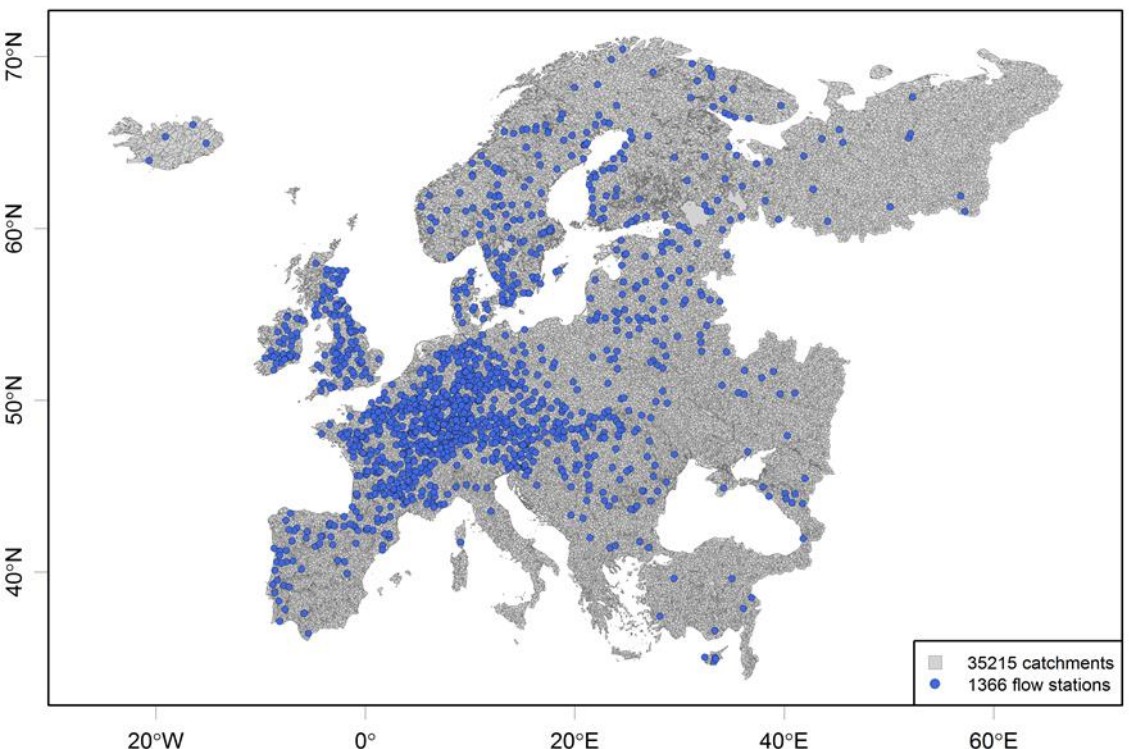

**Figure 1: Spatial extent of the study showing catchments division and selected river gauges.**



**Table 1. Catchment descriptors and the original source of information. Type of descriptor is indicated in brackets after variable name (T=topography; LU=land use; S=soil type; G=geology; C=climate).**

| Variable | Unit | Data source | Description |
|---|---|---|---|
| **Area (T)** | Km$^2$ | SMHI: E-HYPE (Donnelly et al., 2016) http://hypeweb.smhi.se/ | Total upstream area of catchment outlet |
| **meanElev (T)** | m | USGS: Hydrosheds and Hydro 1K (for latitude >60°) (Lehner et al., 2008) | Mean elevation |
| **stdElev (T)** | m | (same as above) | Standard deviation of elevation |
| **meanSlope (T)** | - | (same as above) | Mean slope |
| **Drainage density (T)** | Km$^{-2}$ | (same as above) | $\dfrac{Total\ length\ of\ all\ streams}{Area}$ |
| **10 Land use variables (LU)** | - | CORINE; GLC2000 (Bartholomé et al., 2005) (for areas not covered by CORINE); GGLWD (lake area, distribution, Lehner and Döll, 2004); EIM (EU scale irrigation, Wriedt et al. 2009); GMIA (global scale irrigation, Siebert et al. 2005) | % of catchment area covered by the following land use types: water / glacier / urban / forest / agriculture / pasture / wetland / open with vegetation / open without vegetation / irrigated |
| **7 soil variables (S)** | - | ESD (Panagos 2006); DSMW | % of catchment area covered by the following soil types: coarse soil / medium soil / fine soil / peat / no texture / shallow / moraine |
| **21 geological variables (G)** | - | USGS Geological maps of Europe and the Arabian Peninsula (Pawlewicz et al, 1997, Pollastro et al., 1999) | % of catchment area covered by the following geological classes: Cenozoic (Cz), Cenozoic-Mesozoic (CzMz), Cenozoic-Mesozoic intrusive (CzMzi), Cenozoic volcanic (Czv), Mesozoic (Mz), Mesozoic-Paleozoic (MzPz), Mesozoic-Paleozoic metamorphic (MzPzm), Mesozoic intrusive (Mzi), Mesozoic metamorphic (Mzm), Mesozoic volcanic (Mzv), Paleozoic (Pz), Paleozoic intrusive (Pzi), Paleozoic metamorphic (Pzm), Paleozoic-Precambrian (PzpCm), Paleozoic-Precambrian metamorphic (PzpCmm), Paleozoic volcanic (Pzv), intrusive (i), metamorphic (m), Precambrian (pCm), Precambrian intrusive (pCmi), Precambrian volcanic (pCmv) |
| **Karst (G)** | - | World Map of Carbonate Soil Outcrops V3.0 | % of catchment area marked as "carbonate outcrop" in the World Map of Carbonate Soil Outcrops V3.0 |
| **Pmean (C)** | mm | WFDEI (Weedon et al., 2014) | Mean annual precipitation |
| **SI.Precip (C)** | - | | Seasonality index of precipitation: $$SI = \frac{1}{\bar{R}} \cdot \sum_{n=1}^{12} \left| \bar{x}_n - \frac{\bar{R}}{12} \right|$$ $\bar{x}_n$ : mean rainfall of month $n$, $\bar{R}$ : mean annual rainfall |
| **Tmean (C)** | °C | WFDEI (Weedon et al., 2014) | Mean annual temperature |
| **AI (C)** | - | Precipitation, Temperature and wind from WFDEI (Weedon et al., 2014) | Aridity Index: PET/P where PET is the potential evapotranspiration calculated with Jensen-Haise algorithm (Jensen and Haise, 1963) |



Flow signatures were compiled using daily hydrograph time-series of the Global Runoff data Center (GRDC) and European Water Archive (EWA) databases from 2690 flow gauges across our study domain selected based on agreement between catchment size in metadata and the delineation in E-HYPE (Donnelly et al, 2012). A subsample of this database was selected for this study according to data availability. In order to ensure the reliability of the analyses on flow signatures, only gauging stations with at least five whole calendar years of continuous daily data have been selected (2016 stations). Others subsamples with longer time series (such as 10, 15, 20, 25, and 30 years) were extracted for result evaluation. No missing data was allowed over the period and the longest continuous time-series was used at each gauge. This means that time periods differ between gauging stations but consistent descriptors of precipitation and temperature were always used to match the observed period. Finally, all hydrographs of the resulting subset of flow gauges were visually checked over a 10-year period. This quality assurance mainly eliminated stations with strong flow regulation, obviously erroneous hydrographs or wrong time steps (e.g. monthly). After this selection, the final set of streamflow stations used in the study included 1366 gauging stations.

For the set of river gauges, 16 flow signatures were computed (Table 2). The choice of flow signatures has been guided by a study by Olden and Poff (2003), which provides recommendations for selection of nine indices describing flow regimes with importance to hydro-ecology. In addition, five flow signatures commonly used in hydrology have been added for comparability (Qsp, CVQ, Q5, Q95, RBFlash) and two variables describing catchment response were calculated (RunoffCo and ActET).

**2.2. Cluster analysis for catchment classification**

We classified the catchments based on their similarities in 1) flow signatures for gauged sites only, 2) physiographic descriptors, and 3) physiographic descriptors selected from regression tree analysis on the classes identified using method 1. For the first two analyses, we used the same cluster analysis approach. The catchments were grouped into classes of similar characteristics (of physiography or flow signatures, respectively) using a hierarchical minimum-variance clustering method. The method groups clustering objects (catchments) so that the within class variability is minimized using a combination of the k-means algorithm (Hartigan and Wong, 1979) and Ward's minimum variance method (Ward Jr., 1963). Clustering was started with the k-means algorithm with a large number of classes (50 classes in this work) and classes were merged hierarchically using Ward's minimum variance method. Two classes are merged in such a way that the increase in the sum of the within class variance of the classification variables weighted by the respective class size across all classes is the minimum. After each merging step, the k-means algorithm was applied to the reduced number of classes. The optimum number of classes was established by evaluating the changes in the sum of the weighted variance of the variables across all classes between successive merging steps. The point where the rate of change becomes steeper is set as the optimum number of classes.





**Table 2. Description of the 16 flow signatures studied.**

| Component of flow regime | | variable | Unit | Description |
|---|---|---|---|---|
| **Magnitude of flow events** | **Average flow conditions** | skew | - | skewness = mean/median of daily flows |
| | | Qsp | $L.s^{-1}.km^{-2}$ | mean specific flow |
| | | CVQ | - | coef. of variation = st. deviation / mean of daily flows |
| | **Low flow conditions** | BFI | - | Base flow index: 7-day minimum flow divided by mean annual daily flow averaged across years |
| | | Q5 | $L.s^{-1}.km^{-2}$ | 5th percentile of daily specific flow |
| | **High flow conditions** | HFD | - | High Flow discharge: 10th percentile of daily flow divided by median daily flow |
| | | Q95 | $L.s^{-1}.km^{-2}$ | 95th percentile of daily specific flow |
| **Frequency of flow events** | **Low flow conditions** | LowFr | $year^{-1}$ | total number of low flow spells (threshold equal to 5% of mean daily flow) divided by the record length |
| | **High flow conditions** | HighFrVar | - | coef. of var. in annual number of high flow occurrences (threshold 75th percentile) |
| **duration of flow events** | **Low flow conditions** | LowDurVar | - | coef. of var. in annual mean duration of low flows (threshold 25th percentile) |
| | **High flow conditions** | Mean30dMax | - | mean annual 30-day maximum divided by median flow |
| **timing of flow events** | | Const | - | Constancy of daily flow (see Colwell, 1974) |
| **rate of change in flow events** | | RevVar | - | Coef. of var. in annual nb of reversals (= change of sign in the day-to-day changes time-series) |
| | | RBFlash | - | Richard-Baker flashiness: sum of absolute values of day-to-day changes in mean daily flow divided by the sum of all daily flows |
| **Catchment response** | | RunoffCo | - | Runoff ratio: mean annual flow (in mm/year) divided by mean annual precipitation |
| | | ActET | $mm.year^{-1}$ | Actual evapotranspiration: mean annual precipitation less mean annual flow (in mm/year) |

We performed classification using 16 flow signatures and 35 of the catchment descriptors, which have some correlation to flow signatures (correlation significance tested on Pearson correlation using a t distribution with a threshold of 0.05). In order to reduce the effect of possible correlations between the different physiographic descriptors or flow signatures, we applied principal component analysis (PCA). PCA enables derivation of a set of independent variables, which could be much fewer than the original variables, thereby reducing the dimensionality of the problem. The number of principal components selected for further classification was fixed so that they account for at least 80% of the total variance of the original variables.

The third classification was done for all catchments – both gauged and ungauged, using a predictive regression tree, so called CART (Breiman et al., 1984), calibrated to match the classes identified with method 1. CART stands for Classification And Regression Trees and gathers algorithms based on recursive partitioning, aiming either at classifying a sample or at predicting a dependent variable (here the class of the flow stations classification) based on a set of explanatory variables





(here the set of physiographic variables). The resulting model can be represented as a binary tree: at the different consecutive levels (nodes of the tree), two groups of catchments are divided based on a logical expression using one of the explanatory variables (dominant catchment descriptors). The idea was to obtain a classification close to the one based on the flow signatures but available for the whole set of catchment. Using the CART methodology, a regression tree was first adjusted to

predict the classes of the flow signature classification using criteria based on catchment descriptors, and then this tree was used in a predictive way to classify all catchments in the domain. It was calibrated using an automatic recursive partitioning based on methods described by Breiman et al. (1984) and provided in the R package "rpart" (see Atkinson and Therneau, 2000).

## 2.3. Analysis of physiographic controls of flow characteristics

To examine the link between physiography and flow regime across the geographical domain, matrices of correlation coefficients between all pairs of catchment descriptors and flow signatures were computed using three different correlations: Pearson correlation, Spearman correlation and distance correlation (e.g. Székely and Rizzo, 2009). Catchment descriptors, which did not have any significant relationship with any of the flow signatures, were removed from further analysis. The matrices were accompanied with visual analysis of scatterplots of all pairs of variables for quality control to avoid

disinformation and misunderstanding in the following analysis. Statistical distributions of flow signatures were plotted for different subsets of stream gauges according to the minimum length of the period of continuous daily data availability. Unrealistic values, such as runoff ratios above 1, identified gauging stations that should be further explored and filtered out for the following analyses. Similarly, spatial distributions of all catchment descriptors and flow signatures were plotted as maps. Most of the maps show rather coherent patterns across Europe and could thus be compared to other sources and local

knowledge for further visual quality control.

To evaluate the importance of catchment classification, we compared performance of regression models when calibrated over the whole domain versus those where regressions were derived separately for each class of classified catchments. Multiple regression models were established using a stepwise algorithm for each flow signature as functions of catchment descriptors. This was done for the whole domain and for the three different clustering approaches (above). For a given flow

signature, models were explored using a forward regression, starting from a simple model using only the best correlated descriptor (according to Pearson's linear correlation) and up to a model including all descriptors. At each step, the descriptor giving the best improvement of BIC (Bayesian information criterion) is added, and the algorithm stops when no further improvement can be obtained. The coefficient of determination of each model was then plotted and the final number of variables was determined based on this plot. For a given classification, as many models as the number of classes in the

classification were calibrated for each of the 16 flow signatures and their joint performance were evaluated at the scale of the whole set of stream gauges. To be consistent, regression models were only analyzed for clusters with more than 30 gauging stations, and therefore 17 gauging stations (from 2 classes of the catchment descriptors classification and 1 class of the flow signatures classification) were removed from this analysis because they ended up in classes with fewer stations. In total, 480





regression models were used in the analysis. For each classification method and flow signature, we explored the influence of different physiographic variables by examining the partial correlations of different types of descriptors in the regression.

To gain better understanding of processes behind the hydrologic variability, we examined similarities in both flow signatures and catchment descriptors for each of the clusters based on the CART classification. Each cluster was described by analyzing geographical locations, most characteristic physiography and flow regime. Based on this analysis, hydrological interpretation was used to define potential drivers of hydrological processes, which are dominant in each cluster. The analysis was assisted by several sources of information for classes and sub-classes, such as boxplots of variability in both flow signatures and catchment descriptors, matrices showing the median characteristics in each class, visualization of hydrographs in diagrams, and mapping spatial patterns geographically (most of this material is found in the supplementary material).

## 3 Results and Discussion

### 3.1 Correlation analysis

Significant correlations (significance were tested based on a t distribution with a threshold of 0.05) were found for 400 (out of 786) relations between flow signatures and catchment descriptors from open-data sources (Fig. 2); for instance positive correlation between mean slope and specific flow or low flows, and negative correlation between agricultural area and runoff ratio, and between aridity index and specific flow, 5th and 95th percentiles and runoff ratio. Overall, these relationships seem to be consistent with our a priori knowledge (e.g. Donnelly et al, 2016).

As shown in Fig. 2, there were no big differences between the three types of correlations compared. As expected, more significant correlations appear when using Spearman correlation than Pearson correlation, but the coloring shows that the differences are not very large. Pearson and distance correlation matrices have similar patterns (putting aside that distance correlation cannot be negative). More significant correlations were found when using distance correlation compared to Pearson, but still less than when using Spearman correlation. One exception appears however, which is the percentage of "Cenozoic-Mesozoic igneous" (CzMzi) geological class. This percentage appears to be highly correlated with some of the flow signatures including the skewness of daily flow, the high flow discharge and the mean 30-days maximum. These high correlations (around 0.8) are noticeable in the Pearson (Fig 2a) and distance matrices (Fig. 2c), but absent from the Spearman matrix (Fig 2b). This high correlation led to further examination of the scatterplot between the two variables. It appeared that only a few catchments contain a significant percentage of this geological class and the scatterplot shows that the high correlation is only due to a few points with high values for both "CzMzi" and the concerned flow signature. These high correlations were thus ignored in the following analysis. In addition, combined analysis of the Pearson correlation matrix and the scatterplots showed that a number of catchment descriptors did not have any significant relationship with any of the flow signatures (or only very low correlations – below 0.15). To simplify the process by reducing the number of variables, this led us to remove the following geological catchment descriptors for the rest of the analyses: CzMzi, Czv, i, m, Mzi, Mzm,





MzPz, MzPzm, Mzv, pCmv, PzpCm, Pzv and karst. The low correlation of these variables could be due to small areal

representation in the geographical domain, poor data quality or small influence of subsurface geology on surface hydrology.



**Figure 2. Correlation between catchment descriptors and flow signatures using a) Pearson, b) Spearman and c) distance correlations, respectively. Non-significant correlation according to the significance test are indicated with a dash (-).**

## 3.2 Catchment classifications and regression analysis

A classification based on flow signatures was performed first and we found that 11 classes were optimal for the database

used in this study. The same number of classes was then chosen for the classification based on catchment descriptors. As

described in section 2, the third classification (through CART analysis) was based on the classes from the classification of

flow signatures. However, the class no.2, which contains only 4 gauges (all situated in Cyprus), was excluded from the





CART analysis for consistency purposes. As a result, the classification derived from the CART tree only contains 10 classes (numbered 1, 3-11).

Concerning the CART analysis and classification, we found that 20 nodes in the tree was a good compromise to allow all 10 classes to be predicted while minimizing the complexity of the tree (to make the relationships between catchment descriptors

and signatures interpretable) and while maximizing the probability for correct classification of catchments (relative error=0.59; minimum probability of correctly classified stations at a node = 0.35). The average percentage of correctly classified gauged catchments in each class was 60% (range between 35% and 88% for each leaf node, see Table A in supplementary material). It should be noted that one node (node 3a, see Fig 6) contained more than a third of the catchments (13 645 catchments) and only 35% of the gauges in that node were correctly classified. Efforts to further classify catchments

in this node through an increase of the complexity of the tree did not result in a good compromise. Indeed, to reach a level of 40% of correctly classified gauges at all nodes, the tree had to be detailed up to more than 400 nodes, making any hydrological interpretation of the splits impossible.

The first two classifications, based on clustering of either the flow signatures or the catchment descriptors alone, resulted in very different spatial patterns of similarity across Europe (Fig. 3, note that there is no correspondence between the

numbering of the catchment classes used in maps 3a and 3b). The third classification – where we predict the flow-based classification from the catchment descriptors – exhibits spatial patterns that are rather similar to the flow signatures-based classification, which is expected since the former is derived from the later through a CART predictive regression tree.

In order to analyze the specific characteristics of the different clusters in terms of catchment descriptors and flow signatures, boxplots representing the distribution of each variable within the clusters were plotted (see sections C.1 and C.2 of the

supplementary material). For the classification based on flow signatures (Fig. 3a), some clear distinctions appear between clusters in terms of mean specific flow and coefficient of variation of daily flow. For example, clusters no. 7 and 10 have the highest mean specific flows while clusters no. 2 and 4 have the highest coefficients of variation. Concerning percentage of agricultural area, some clusters cover a wide range of values (no. 3, 4, 5, 11) while others contain mostly catchments with low percentages of area covered by agriculture (no. 1, 7).

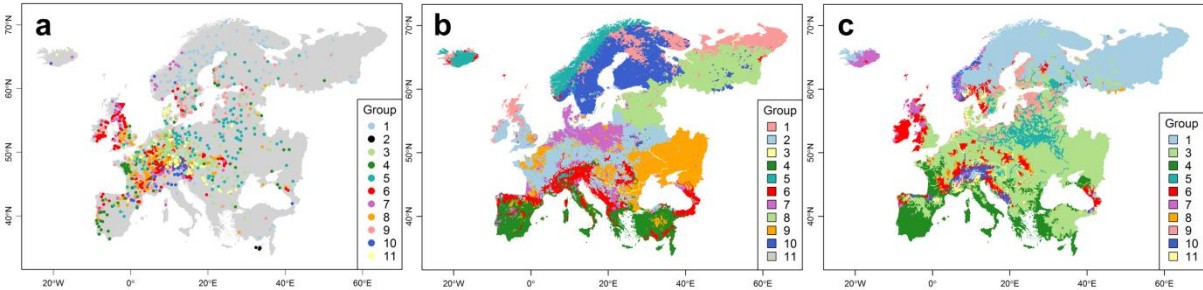

**Figure 3. Spatial patterns of catchment classification across Europe based on a) flow signatures at flow gauges, b) catchment descriptors, and c) CART predictive regression tree.**



The spatial pattern in Figure 3b (based on catchment descriptors) show geographically coherent patterns with for example cluster No 6 bringing together mainly mountainous areas, No 4 gathering southern warm catchments , No 7 representing plain regions of the Netherlands, northern Germany, Denmark and Poland. Analysis of the distribution of the different

variables in the classes (see boxplots in section C.2 of the supplementary material) showed for example that cluster No 5, which is mainly located in Western Norway and Iceland, gathers catchments with low mean temperatures and high mean precipitations with high proportion of open areas without vegetation. In terms of flow signatures, these catchments have high mean and high flows, high runoff ratios and low evapotranspiration. Cluster No 11contains 323 catchments but none of them correspond to a stream gauge included in the study. Thus, no observations are available to characterize flow signatures for

this class. Observations are limited as well for cluster No 3 as only 13 of the 152 catchments that belong to this class correspond to a flow station. These two classes were thus excluded from further analysis.

Only clustering using catchment descriptors or CART can be applied for the whole domain, i.e. in ungauged catchments. The CART-based catchment classification (Fig. 3c) was chosen for more detailed analysis (in Section 3.3) on similarities in flow generation processes as the clusters were more homogenous. Average of standard deviation within all clusters was estimated

to be 0.71 for catchment descriptors and 0.78 for flow signatures using catchment descriptors for classification, while it was 0.76 for catchment descriptors and 0.67 for flow signatures using CART. Hence, the former discriminates classes more in terms of physiography (0.71 vs 0.76 for the CART classification) and the CART classification discriminates classes more in terms of flow signatures (0.67 vs 0.78).

Figure 4 shows that catchment classification did improve the overall prediction of flow signatures using regression models

across the whole European domain. This could be expected as using 10 models instead of one increases the degree of freedom as the number of calibrated parameters increases. On average, classification using catchment descriptors and CART improved the model performance by 14.7% and flow signatures by 33%. The latter yields the best results since this classification is based directly on the discriminating variables (flow signatures). There are few differences in terms of the performance of the models obtained using either the catchment descriptors or CART for classification, the later giving

slightly better results for most of the variables (e.g. Q5, High Flow Discharge, high flow frequency variability, variability of reversals, flashiness, runoff ratio), but poorer results for base flow index and low flow frequency. The performance of the regression models for the different flow signatures will be further discussed in part 3.4.





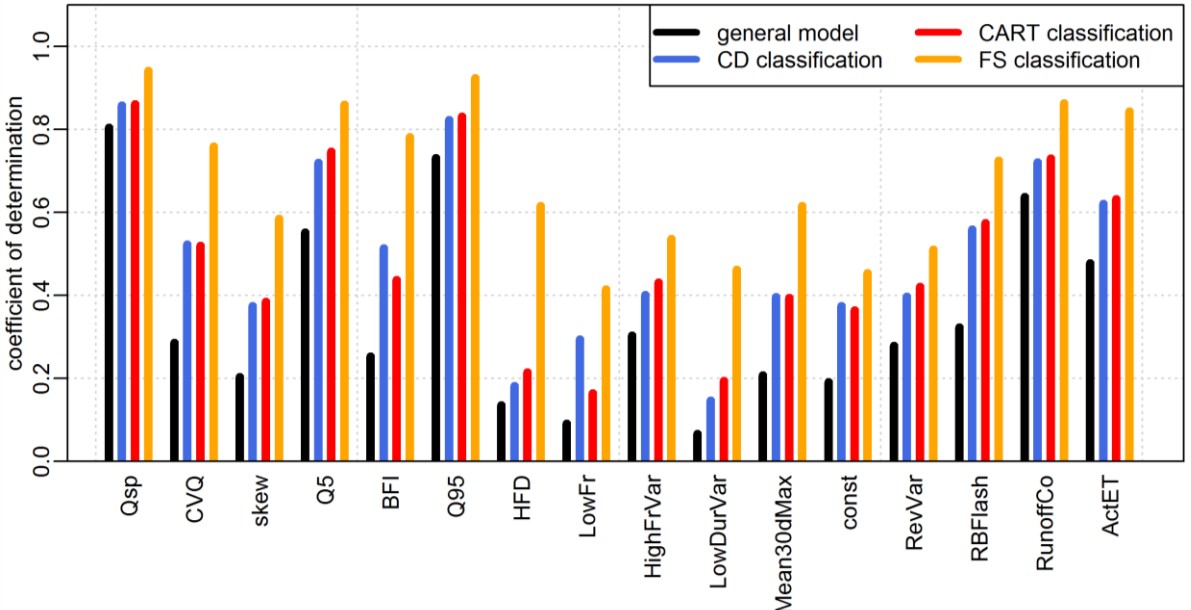

**Figure 4. Performance of regression models when calibrated for each flow signature (Table 2) and applied over the whole domain with a general model or one per class, using catchment classification based on catchment descriptors (CD), flow signatures (FS) or regression tree (CART). Performance is evaluated over the whole set of flow gauges together even if different models are used in different classes.**

The partial correlation analysis of the regression models shows that there are different controls for the different flow signatures (Fig. 5). The highlighted controls are rather similar for the different classification methods. Climatic descriptors play the most important role for most of the flow signatures, especially those related to average and high flows. For the base-flow index, geology is more important and for the flashiness of flow, topography is the main control. Topography also plays an important role in low flows magnitude (Q5), being the main driver for this signature in some of the classes and for the global model. For most of the flow signatures, the second most important descriptor is generally land use (mean flow, high flows, runoff coefficient, ET, variability of reversals).

The importance of the different controls varies across the classes (length of the boxplots in Fig. 5) and the main drivers for a given variable can also differ between classes (not shown in the figure). As an example, for evapotranspiration, land use is the main driver in classes 7, 8, 10 and 11 of the CART classification while climate plays the most important role in the other classes. Runoff ratio is mostly explained by soil type in class 11, by land use in classes 6 and 10 and by climate in the other classes (see detailed analysis in Section 3.3). High flow magnitude, also mainly driven by climate in most of the classes, is explained by topography in class 11 and by land use in class 5. It is interesting to note that climate is a strong driver for almost all signatures in class 4 (warm regions in southern Europe) while other drivers play an important role in other parts of Europe, for example in class 7 (topography, land use and geology are important), 9 (topography) 10 (topography and land use).





**Figure 5. Partial $R^2$ of different type of descriptors (Table 1) used in the regression models for flow signatures (partial $R^2$ for the type of descriptors is the sum of partial $R^2$ of variables from that type used in the regression model). The boxplots show the range of values among the models calibrated in the different classes using the different catchment classification methods: a) flow signatures at flow gauges, b) catchment descriptors, and c) CART predictive regression tree. The black point gives the value for the general model calibrated over the whole domain.**

The identified controls for the different flow signatures are generally consistent with the findings of several recent studies conducted in different parts of the world. For instance, Longobardi and Villani (2008) and Bloomeld et al (2009) found a strong relationship between the base flow index and geology for the Mediterranean area and the Thames basin, respectively. Similarly, Holko et al (2011) found out that flashiness index is correlated with geology, catchment area and elevation as well as percentages of agricultural and forest landuses for catchments in Austria and Slovakia. For catchments across the US,



Yaeger et al (2012) found out that the upper tail of the flow duration curve is controlled more by precipitation intensity while the lower tail is more controlled by catchment landscape properties, such as soils, geology, etc. For the same US dataset, Sawciz et al. showed that runoff coefficient was dominated by aridity of the climate, and that baseflow index was controlled by soil and geological characteristics. The influence of topography on the magnitude of low flow was also found by

Donnelly et al (2016) through a correlation analysis of a set of flow signatures and catchment descriptors across Europe.

## 3.3 Hydrological interpretation using CART

The regression tree classification (CART) enabled a better understanding of the main controls driving the separation into classes, as the classes of flow signature combinations are predicted from the available catchment descriptors. In the resulting tree (Fig. 6), the main variable separating the different classes is the Aridity Index (AI) with a separating value close to 1.

This value separates the energy-limited catchments (AI<1) from the moisture-limited catchment (AI>1). Mean temperature is the second separating variable; followed by variables describing soil types (peat, moraine), land use (agriculture, open without vegetation, wetland, forest), topography (area, mean elevation) and climate (precipitation seasonality index, mean precipitation). This indicates the order of importance of catchment descriptors that control flow signatures.

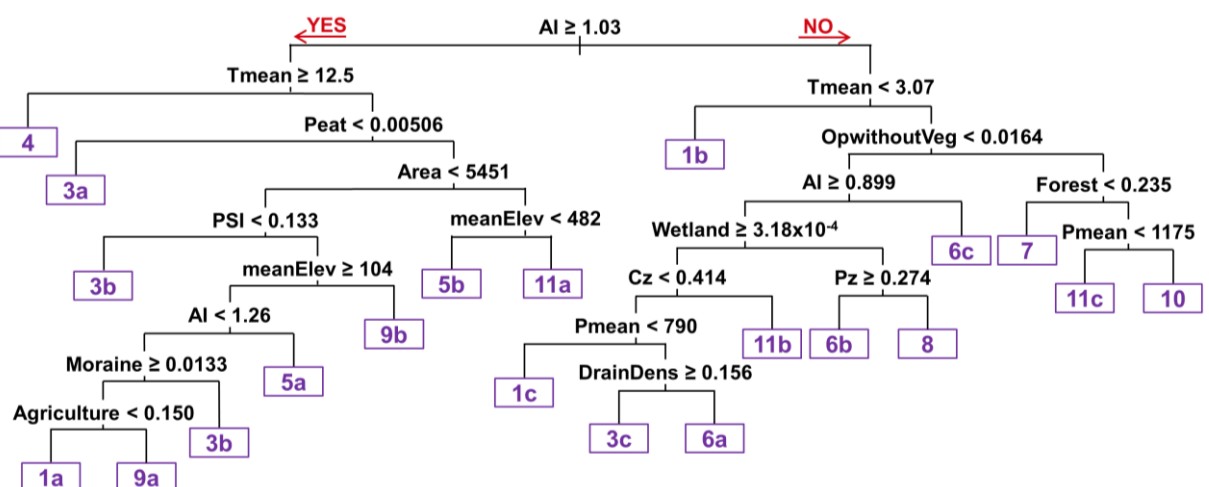

**Figure 6: CART tree adjusted on the FS classification and used as a predictive tree for the "CART" classification.**

Some of the differences between the hydrographs within catchment classes and across catchment classes can be seen in Fig.7, where we show examples of the observed time-series. We found the following characteristics, which are summarized in Table 3 and further supported by results figures in section B of the supplementary material:

*Class 1* has a rather smooth flow, seasonal flow pattern with a very pronounced spring flood peak. This is a cold northern part of Europe and some parts of the Alps and Caucasus, characterized by spring snowmelt with some dampening in lakes and wetlands.





*Class 3* is very big (about 1/3 of the catchments) miscellaneous class without any distinct character. As explained in section 3.2, efforts to further classify catchments in this class (and more specifically in node 3a) did not succeed.

*Class 4* is characterized by very spiky hydrographs with high peaks and low baseflow. The flow regime is high winter flows and low summer flows. This is the Mediterranean area characterized by arid climate, flow seasonality and human impacts.

*Class 5* shows overall low flow with some influence of snow melt (spring flood) for some catchments during some years. This is Northern part of central-eastern Europe characterized by low flashiness due to the large amount of water bodies, low slopes and low elevation, which dampen the flow.

*Class 6* has very high peaks especially during winter and high flow in general. Overall, flashy flow with a tendency to lower flow during summer and geographically scattered humid areas all over Europe.

*Class 7* shows in general high and flashy flow, for most catchments higher winter flows but for some catchments summer high flows instead due to snow and glaciers melt (this is the class with most glaciers, see  Fig. F of the supplementary material). This class encompasses wet and cold mountainous areas along the coasts in north-western Europe and some humid parts of the Alps.

*Class 8* is characterized by peaky flow all over the year with higher peaks in winter. This class consists of smaller headwater
catchments in some warm and humid parts of central, south-western and north-eastern Europe.

*Class 9* has rather low flow with a snow melt dominated spring flood. Low amplitude but frequent short term variability. This is mainly flat lands around the Baltic and Northern Sea characterized by forests, lakes and wetlands. Some catchments are characterized by similar geological structures (Pz, pCmi, see Fig. J in the supplementary material).

*Class 10* shows high flows with very high and frequent peaks, some tendency to spring peak but also high flow during
winter. Frequent short-term variability is common in these wet, high elevation and steep catchments across mountain ranges of Europe.

*Class 11* is characterized by sustained flow with high baseflow and some tendency to spring peak in some catchments but overall low seasonality of flow. These catchments are below mountains and in lower parts of large river basins. We suspect some artifacts in the classification when extrapolating the CART tree to the full European domain, as parts of the catchments
in this class were not representative to the river gauges in the same class (see Fig. L in the supplementary material, showing that the gauged catchments in node 11b do not have the same characteristics as the rest of the catchments in the same node).





**Figure 7. 3-years hydrographs (left) and average annual hydrographs based on > 5 years daily flows (right) at the stream gauges of the CART classification classes. Grey->black: all stream gauges belonging to the class; red: stream gauge where the flow signatures are closest to the class median flow signatures. Note that the scales are different for classes 5 and 9 and that this classification doesn't contain any class no. 2 as explained in part 3.2.**





**Table 3. Summary of findings when using the CART tree to classify catchments (CART classification shown in Fig. 3c) and extracting the main features for each cluster. Appointed flow signatures (Table 1) and catchment descriptors (Table 2) have median values in the 30% low/high percentile of the distribution over whole domain. Bold indicates median values in the 10% low/high percentile. Supporting figures with boxplots and matrices of flow signatures and catchment descriptors as well as detailed maps of spatial patterns are found in the supplementary material (resp. in sections B.1, B.2 and B.3).**

| Class | Sample size | | Flow signatures (FS) | | Catchment descriptors (CD) | | Spatial Pattern | Dominant hydrological processes |
|---|---|---|---|---|---|---|---|---|
| | No. of catchm. | No. of gauges | FS low | FS high | CD low | CD high | | |
| 1 | 6878 | 112 | RBFlash, **ActET** | RunoffCo, HighFrVar, Mean30dMax RevVar | Urban, **Agriculture**, Pasture, Medium AI, DrainDens, Pmean, **Tmean**, | **Water**, Forest, **Wetland**, OpwithVeg,, Peat, **NoTexture, Moraine**, PSI, pCm, PzpCmm | N and center Scandinavia, W Iceland, Russia | Snow dominated flow regime with significant snow melt during spring but rather even flow during the rest of the year due to dampening in lakes, wetlands and low evapotranspiration. Flow influenced by some hydropower regulation. |
| 2 | - | - | | | - | - | - | - |
| 3 | 14282 | 536 | - | - | - | Agriculture, Moraine, PzpCmm | Large coverage in Western, Central and Eastern Europe. | - |
| 4 | 5112 | 91 | Qsp, Q RunoffCo, **BFI** | **CVQ**, const, RBFlash, HFD, LowFr, **skew**, **Mean30dMax** | Forest, Pasture | Agriculture, **Irrigated**, Moraine, **Tmean**, PSI, **AI**, PzpCmm | Southern and Eastern part of Europe | High ET and high human alteration of natural processes. Winter flow is dominated by precipitation while summer flow is limited by evapotranspiration. |
| 5 | 1765 | 72 | Qsp, CVQ Q95, **RBFlash** RunoffCo, skew, HFI Mean30dMax | BFI, HighFrVar, LowDurVar, RevVar | meanElev, stdElev, meanSlope, Pmean | area, Water, Agriculture, Coarse, Peat, Moraine , AI, Cz, PzpCmm | Mainly Poland, Belarus, Lituania, some in S Sweden and Russia | Water flow is dampened by large river channels and water bodies and flat lands. Some influence of snowmelt driven flows. One sub-class (5b) is more controlled by water bodies and the other (5a) by surrounding flood plains. |
| 6 | 3325 | 261 | HighFrVar, RevVar | Qsp, Q95, RBFlash, RunoffCo | AI | Pasture, Moraine, Pmean, PzpCmm, | Rather scattered distribution: the Brittish Islands, S. Scandinavia, Russia, lower regions of mountainous areas. | Precipitation driven frequent peak flows. One sub-class with rapid response due small area and high slope (6b). |



| | | | | | | | | |
|---|---|---|---|---|---|---|---|---|
| 7 | 678 | 33 | **ActET**, HighFrVar, LowDurVar, RevVar | **Qsp, Q5, Q95**, RBFlash, **RunoffCo** | Urban, Forest, **Agriculture,** Medium , DrainDens, Tmean, **AI** | stdElev, meanSlope, Wetland, Peat, **OpwithVeg, Pmean, OpwithoutVeg**, NoTexture , Shallow, Moraine, PzpCmm | SE Iceland, Scotland, W Norway, some in the Alps | Low storage (in soil and water bodies) that generates quick response to rainfall. Most catchments have rainfall dominated flow but also some are snow and glaciers melt dominated. |
| 8 | 670 | 63 | BFI, HighFrVar | CVQ, RBFlash, ActET, skew, LowFr, | area, OpwithVeg, NoTexture | Pasture, Moraine, Pmean, Tmean, Mz, PzpCmm, | Close to class 6 regions in center of France, Carpathians and Russia | Fast response to precipitation since they are small headwater catchments with low storage capacity. |
| 9 | 969 | 52 | Q5, RBFlash ActET | HFD, LowFr, LowDurVar, Mean30dMax, RevVar | **meanElev,**, ,stdElev, **meanSlope,** Pasture, Pmean, Tmean | Water, Forest, Wetland, Peat, NoTexture, Moraine, PzpCmm, | Around Baltic Sea and along the Northern Sea and English Channel coast. | Snow dominated flow regime with significant snow melt during spring. Indications of short-term regulations. Continuous contribution through lateral flow leading to a more sustained flow. |
| 10 | 762 | 79 | CVQ, skew HFD, HighFrVar, Mean30dMax RevVar | **Qsp, Q5**, Q95, RunoffCo, BFI, const | Agriculture, Tmean, **AI** | **meanElev, stdElev , meanSlope, Pmean,** OpwithVeg, PzpCmm, **OpwithoutVeg**, Shallow, Moraine, | Mountainous regions of W Norway, Pyreneous, Alps, Bosnia, Montenegro, few in Carpathians and Scotland | Regulated flow for hydropower production during winter but still with some tendency of spring flow. |
| 11 | 774 | 67 | **CVQ** , RBFlash **skew , HFI Mean30dMax** | Q5, BFI | - | area, meanElev, **stdElev,** meanSlope , Water, Irrigated, OpwithoutVeg, Coarse, Moraine, DrainDens, Cz, pCm, Pzi, PzpCmm | SE France, NE Italy, W Danmark, SE Norway, some in Sweden, large catchments of big rivers like Rhine and Danube | Flow is governed by continuous supply from upstream storages either from large upstream areas or upstream mountains. (Note: Some catchments (e.g in Denmark) are not representative to the gauges in this class) |





The hydrological interpretations of the detected spatial patterns (Table 3) pointed to climatology as the main control of the hydrological processes in most classes (which is consistent with AI as the main control in Fig. 6). This is highlighted by the notable influence of rainfall-driven river flow in clusters No 6, 7, 8 (Western and Northern Europe) throughout the year, and during winter in 4 (Southern and Eastern Europe). The latter region is most obviously strongly affected by evapotranspiration, while snow-dominated regimes with a spring melt season are characteristic for clusters No 1, 7, 9 and to some extent also No 5 and 10. These clusters are found in the Northern and mountainous parts of Europe.

Regarding landscape influence, dampening effects of river flow response are found in clusters No 1 and 5, due to passage through many waterbodies and vast flatlands. Continuous supply to river flow is found in cluster No 9 and 11 through lateral flow, large contributing area or upstream mountainous areas. On the other hand, clusters No 7, 8 and 6b show fast response and low storage capacity, which could be attributed to thin soils, high slopes or small catchments.

Impact from hydropower production was found in clusters No 1, 9, 10, which were all snow dominated but showed redistribution of water during the year due to regulation and in some cases influence of short-term regulation. It should be noted that this effect was visible although the gauges from most regulated rivers were already excluded from the study (section 2.1). Human alteration was also assumed to dominate the hydrological processes in cluster No 4, where the hydrographs did not look natural and irrigation is high (Southern and Eastern Europe).

Some clusters were found to have similar flow signatures for different reasons. For instance, the damping of peak flow in cluster No 5 could be caused by either water bodies (5b) or floodplains with a wider river channel (5a). Some clusters were easier to distinguish with many different characteristic signatures or physiography (e.g No 1, 4, 7, 10), while others do not have particular signatures that stand out in terms of their magnitude from the rest (e.g. No 6 and 8). Again, it should be recognized that 1/3 of the catchments (the class no. 3) could not be interpreted hydrologically as they did not show similarities in flow signature values and shared only few catchment descriptors (within 30% percentile of agriculture, moraine and one geological feature, see Table 3).

Previous studies have noted that large-scale databases are connected with uncertainties and may sometimes even be disinformative at high resolution (Donnelly et al., 2012; Kauffeldt et al., 2013), which may be a reason for some weak statistical relationships and difficulties in catchment classification. European hydrology is also very much affected by human alteration, which is probably not fully covered by the descriptors. Hence, there is still need for further investigations to better understand hydrologic variability across Europe.

### 3.4 Application of the results: predicting flow signatures over Europe

Figure 8 shows the result of predicted flow signatures using the regression models calibrated within each class of the CART classification. As shown in Figure 4, the performances of these models are diverse: some flow signatures are well modelled ($R^2$ above 0.8 for mean specific flow and 95$^{th}$ quantile, above 0.7 for 5$^{th}$ quantile, runoff ratio, skewness of daily flow, mean 30-days maximum), but some other models perform very poorly ($R^2$ below 0.2 for low flow frequency and variability of low flow duration).





The performances also vary from class to class (not shown here). Models are generally poor (most $R^2$ below 0.4, a few between 0.4 and 0.6) in class 3, which is a very large and miscellaneous class, but also for classes 6 and 8. On the other hand, the best performances are observed in classes 7, 10 and 11.

Figure 8 shows that some negative values appear when applying the calibrated regression models to predict flow signatures. This is explained by the larger range of values of the predicting variables in the whole domain than in the subset of 1366 catchments with flow stations. For example, the predicted values for the $5^{th}$ quantile of daily flow are negative in 2607 catchments (over the 35215 modelled), most of them belonging to classes 3 and 4. In class 4, the regression for Q5 uses percentage of forest (positive coefficient) and mean temperature (negative coefficient) as the first two predictors. Some negative values appear when the model is applied to catchments with a low percentage of forest and a high mean temperature.

These mitigated results emphasize the experimental nature of these regression models and that they should not be applied outside of the observed ranges of catchment descriptors. However, these regression models help us improving our understanding of European hydrological processes and identifying the dominant controls of the flow signatures in different parts of Europe (see section 3.2). This understanding can be useful when building models that include physical reasoning.

One implication of the identified spatial pattern of flow characteristics and their dominant physiographic controls is that one can delineate regions of particular flow characteristics, for which part of the hydrograph is important. This could be related to the season or component of the hydrograph where the flow is more sensitive to the controlling physiographic attributes. In addition to establishing empirical relationships between the flow signatures and catchment physiographic attributes, like we did in this work, this has a potential application in improving dynamical rainfall runoff models across Europe. Design and results of process-based models should be coherent to empirical findings and when applied on the large-scale, they should thus be evaluated against empirical observations of large-scale spatial patterns, like the ones we provided in this paper.

Furthermore, our results can be applied to directly improve process-based hydrological models. We showed that model predictions are improved by 15% when establishing models for separate classes of catchment with similar signatures and controls (see section 3.2). This knowledge is highly valuable when estimating parameter values for continental-scale hydrological models. Currently, there is an emerging need for parameter estimation also in ungauged basins from several modelling communities. For instance traditional catchment models have recently been applied on a pan-European scale, e.g. SWAT (Abbaspour et al., 2015) and HYPE (Donnelly et al, 2016). Accordingly, global hydrological models are starting to develop rigorous calibration procedures (e.g. Müller Schmied et al., 2014). The new empirical knowledge we gained in this work could, for instance, be incorporated in the modelled processes of such models. Processes that control the part of the hydrograph that is sensitive to given physiographic attributes can be parameterized and calibrated as functions of the physiographic attributes for the different catchment classes separately (Hundecha et al., 2016). This will ultimately improve the dynamic models predictive ability, while at the same time enabling prediction in ungauged catchments since the model parameters are functions of the controlling catchment physiographic attributes instead of gauged flow.





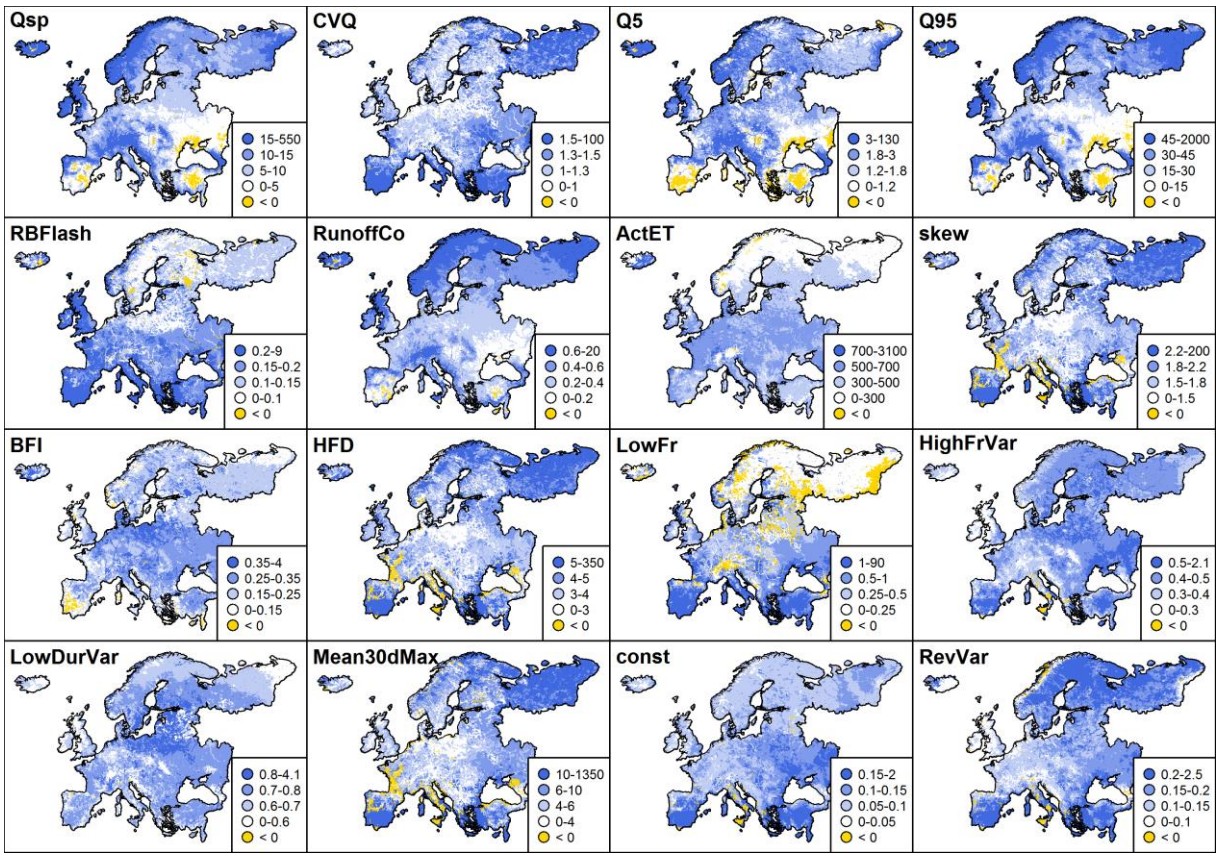

**Figure 8. Predicted flow signatures using the regression models calibrated within classes of the CART classification (Fig 3c). Note that the color intervals are adapted to each signature and do not have a constant size for a given signature: for a better readability they are based on the quartiles of the signature distribution. The coefficients of determination of these models are shown in Figure 4.**

## Conclusions

We set out to better understand hydrological patterns across the European continent by exploring similarities in flow signatures and physiography. Using open datasets and statistical analysis we found it possible to attribute dominant flow generating processes to specific geographical domains. From the analysis of catchment classification using similarities in 16 flow signatures and 35 catchment descriptors across Europe, we can conclude that:

- Physiography is significantly correlated to flow signatures at this large scale and catchment classification improves predictions of hydrologic variability across Europe. Different physiographical variables control different flow signatures; climatic variables play the most important role for most of the flow signatures but topography is more important for flashiness and low flow magnitude while geology is the main control for base flow index.





- Different classification methods (e.g. based on physiographic characteristics *vs* flow signatures) can lead to very different patterns, emphasizing the importance of the choice of the methodology according to the use of the classification. However, classes obtained based on flow signatures can be predicted using physiographic characteristics with on average 60% correctly classified catchments in each class. In total, European catchments can be described through ten classes with both similar flow signatures and physiography. The most important physiography for predicting classes is the aridity index (AI), which separates the energy-limited catchments from the moisture-limited catchments. Thereafter follows variables describing soil types, land use, topography and other aspects of the climate.

- Interpretation of dominant flow-generating processes and catchment behavior (such as rainfall response, snow-melt, evapotranspiration, dampening, storage capacity, human alterations) could explain the hydrologic variability across Europe to a large extent. However, flow signatures from 1/3 of the catchments were not possible to classify or understand based on the physiographical variables used in this analysis. This calls for more detailed analysis with additional data in these areas.

- The links we found between the flow characteristics and physiography could potentially be used in spatial mapping of flow signatures (for instance mean specific flow, 5$^{th}$ and 95$^{th}$ quantiles, runoff ratio, skewness of daily flow, mean 30-days maximum) also for ungauged basins. Moreover, the findings have potential to constrain and derive parameters for process-based models to increase predictability in dynamic modelling. Using many gauges from catchments with similar dominant processes for flow generation gives more robust parameter values, so therefore, the ten classes of similar catchments may facilitate model parameter estimation in pan-European hydrological models.

- Open data sources enable new forms of comparative science and show large potential for research to generate new knowledge and hydrological insights encompassing variable environmental conditions. However, for Europe there is a lack of homogenous datasets for human impact on flows, such as local water management, abstractions and regulation schemes. There is thus still a need for opening up more public sector data for re-use and, especially, for compiling large-scale databases on the global or continental scales across administrative boarders.

**Acknowledgment**

This study was performed within the EU FP7-funded project SWITCH-ON (grant agreement No 603587), which explore the potential of Open Data for comparative hydrology and collaborative research, as well as promote Open Science for transparency and reproducibility. All data, scripts and protocols are available in the SWITCH-ON Virtual Water-Science



Laboratory at http://www.water-switch-on.eu for review. We would like to thank the Global Runoff Data Center (GRDC) for compiling, maintaining and sharing time-series of river flow as monitored by national institutes. Much of the data used in this study was available from input files of the E-HYPE model; the authors therefore also wish to thank staff at the Hydrological Research unit at SMHI for previous efforts on data compilation, especially we would like to acknowledge the
work by Kristina Isberg and Jörgen Rosberg.

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
