# Peer review of "Understanding Hydrologic Variability across Europe through Catchment Classification"

_Hydrology and Earth System Sciences, 2016_

## Referee Comment (RC1) · Anonymous Referee #1 · 13 Oct 2016

This paper uses established methods to classify regions with similar physiographic characteristics and with similar flow signatures to determine the best predictive relations at ungauged locations. For this reason, the manuscript reads more as a report rather than a novel contribution to the literature. For example, in lines 13-15, it seems the manuscript goals do not seem to be driven by scientific hypothesis but more by having a large set of data and wanting to develop/explore some relations which may (or may not) be useful at some later point. In this way, I think the motivation for the study seems weak as a scientific contribution. Despite this, I do believe that the novelty of the manuscript is in the application of these methods over such a large spatial domain. As hydrologic modeling efforts expand to cover continental scales, the ability to upscale existing approaches for model calibration across large ungauged regions becomes a limiting factor in these efforts. This point should emphasized more in the

manuscript to elevate the impact of the work beyond an application of existing methods to a larger region than had been tested previously.

1. The selection of flow signatures needs more details as to how they were selected. Olden and Poff (2003) do not from my remembering of the paper - as the authors indicate in line 16 - provide 9 signatures. Their paper attempts to reduce redundancy in the 200+ statistics that have been used for hydro-ecologic classification but they do not provide a definitive reduced list. More details need to be provided as to why these signatures were selected, particularly because their usefulness in applications is not part of the analysis in the paper. This reads as quite an arbitrary choice.

2. In lines 12-13, the comment is made that this type of analysis has not been applied at the continental scale "including large rivers with human alteration. . ." Do the catchments examined here have human alteration? This is not noted in the methods? Does this bias your results?

3. In line 18, the statement is made that "identified gauging stations that should be further explored and filtered out. . ." Was this actually done?

4. In Section 2.3, how were variables determined to be significant in the regressions? What diagnostics were used? How many variables were allowed to enter in each equation? It may be useful as an explanatory tool to see which variables are significant but to make predictions (which is the goal of this work), one needs to adhere to good statistical practices. How were these practices followed?

5. I found myself questioning the value of Section 3.1. I do not think this offers any additional information beyond what can be determined from the CART and regression analyses. This section also contributes to the manuscript reading more as a report as this section seems to explain what could be characterized as exploratory data analysis that is completed before one settles on an approach and hypothesis to test. I also think that the manuscript is a bit laborious in its reading and removal of this section would help streamline the manuscript.

6. Section 3.2 seems to be missing a reference to how the classification was applied to the data. At the very least, reference Section 2.2 to describe how the classification was completed.

7. I may have missed this but I think it is necessary to develop regression on the flow signatures using the entire dataset to compare to the regression results obtained for the classes. This analysis would determine the objective improvements provided by first classifying the data. If this analysis has been completed, please refer to this in the text when discussing the results.

8. There are two papers that I direct the authors to for potential citation.

Singh et al. (2014) used CART to classify model parameter behavior across the United States and may be helpful to motivate some of other contexts in which CART has been utilized for model parameterization at ungauged locations.

Oudin et al. (2010) ask almost the same question as this paper in how physiographic similarity is related to hydrologic similarity, although they answer this question using actual model results.

Oudin, L., A. Kay, V. Andréassian, and C. Perrin (2010), Are seemingly physically similar catchments truly hydrologically similar? Water Resour. Res., 46, W11558, doi:10.1029/2009WR008887.

Singh, R., S.A. Archfield, and T. Wagener, 2014, Identifying dominant controls on hydrologic model parameter transfer from gauged to ungauged basins - a comparative hydrology approach, Journal of Hydrology, doi: 10.1016/j.jhydrol.2014.06.030, 2014.

---

## Referee Comment (RC2) · M.C. Westhoff (Referee) · 14 Oct 2016

This paper aims to classify a large set of European catchments using a few different regression, and clustering techniques. The results are analyzed by looking at spatial patterns while the main drivers are characterized for each class.

Although I personally have no record in catchment classification methods, I judge this paper as potentially publishable. But before that, I think the paper can and have to be improved.

The first point I was triggered about was the sentence "So far we have not yet found a widely accepted classification system" (P2, L8), which made me expect that this paper would (or at least aimed) to finalize this issue. However, this is not the case, while I think you can make this attempt by reserving a part of the available dataset

for validation. The used dataset is large enough and I think the results would benefit from a "calibration-validation cycle" in which the dataset is split in two randomly chosen sets, of which one is used for calibration and the other for validation. This can be done several times for different randomly chosen subsets. This exercise may tell you more about number of catchments needed in a class and how robust the chosen signatures are.

A second aspect was that I had problems understanding what was done and in which order. If I am not mistaken, I think you can roughly summarize it by: 1) With a regression analysis catchment descriptors (CD) are correlated with flow signatures (FS). 2) Classes have been derived using 3 different clustering methods: one using CD, one using FS and one using a CART analysis. 3) For each class, correlations between CD and FS are derived and compared with the correlations derived in step 1. If this is indeed the case, I suggest to add e.g. a flow chart and to turn paragraph 2.2 and 2.3 around.

I very much agree with paragraph 3.4 in which it is suggested that the finding can be used for ungauged basins or to parameterize large scale models. But to really benefit from the results of this paper I would encourage the authors to also publish the regression constants. This would make it possible for others to indeed parameterize large scale models, while other future classification studies can better compare (quantitatively) their results with those of this study.

Minor comments: Be consistent in using either the term "Catchment Descriptors" or "physiographic control"

P3,L32: Give also the range of the catchment sizes

P6,L5: explain what E-HYPE is

On P3,L11-12 it is stated that "No study so far, to our knowledge, has applied the results from comparative hydrology at the continental scale, also including large rivers

with human alteration and ungauged basins", suggesting that this study will include basin subject to human alteration. Now on P6,L12 it is stated that stations with strong flow regulations were eliminated.

P12,L15: It is unclear to me to which method is referred here. Please clarify

P12,L8: You mean actual evaporation, right? Also add this at P13,L15 and potential other locations.

P14,L9: Is it possible to quantify the strong relationship?

---

## Referee Comment (RC3) · Anonymous Referee #3 · 27 Oct 2016

This is a potentially interesting paper, but needs improvement in the way its methods and results are presented. My main points are as follows:

(1) The purpose of conducting a correlation analysis seems a bit unclear to me. Firstly, if reducing the number of variables (to be used for classification) was the goal, why is it that only physical descriptors were chosen for culling, and not flow signatures? It could easily be argued that some flow signatures (e.g., HFD, LowFr) which do not have high correlation with most physical descriptors can be removed as well. Secondly, as mentioned in Section 2.2, a PCA is performed anyways to reduce the dimensionality prior to classification. So why prescreen the variables with correlation analysis before applying PCA? Wouldn't PCA alone on the whole dataset (16 flow and 48 physical variables) do the job?

(2) In Section 3.2, the geographical patterns of classification are briefly mentioned for the physical descriptors based classification, and not at all for the flow signatures based one. I think the authors have a huge opportunity here to explain the geographical context of the spatial patterns observed in Fig 3a and b. It is mentioned (Page 11, Lines 13 and 14) that the flow and physical descriptor based classifications lead to different patterns. Why is that? Any speculation on this aspect would be quite helpful here because it directly relates to the main questions asked in this study.

(3) It might be helpful to state the proportion of total area covered by each of the 10 classes obtained through CART (Figure 3c). It is mentioned later in Section 3.4 that the regression models used for predicting flow signatures across Europe perform poorly for classes 3, 6 and 8, and perform best for classes 7, 10 and 11. Knowing the % area of Europe covered by poor and good performing classes would clarify the ability of your classification to predict flow signatures in ungauged catchments. Based on a quick look of Figure 3c, it seems to me that your best performing classes are predominantly clustered around the Alps, and majority of the Europe is covered by the poor performing class 3 (and class 6 covers large areas too). Does this mean that after going through all the efforts of two classifications + CART + regression models, our ability to predict flow signatures at ungauged catchments is only limited to wet, mountainous systems (which we already know from previous studies to be simple and easily predictable hydrological systems)?
* * *

---

## Editor Comment (EC1) · B. Schaefli (Editor) · 28 Oct 2016

The paper received three detailed reviews, which all conclude that the paper could become an interesting contribution to HESS but that it requires major modifications. I therefore invite the authors to answer these comments as soon as possible in the public discussion, before preparing a revised version.

---

## Author Response (AR1)

**Point-by-point response to the reviewers with corresponding relevant changes**

As the three reviews pointed out that the goals of our study could be misunderstood, we have reworked different parts of the introduction and conclusion to better emphasize our aim to gain better understanding about hydrological signature patterns and their controls across Europe. We have also tried to lighten and rephrase different parts of the manuscript to make it more straightforward.

The section 3.1 has been moved to the supplementary material, responding to both reviewers 1 and 3. Section 3.2 ("Catchment classifications and regression analysis") has been divided in two sections ("Catchment classifications" and "Using regression analysis to understand controls on individual signatures"), in order to make the paper easier to read and to highlight our motivation of understanding controls. The usefulness of the regression models for reaching this understanding has been further explained in the introduction of the new section 3.2 ("Using regression analysis to understand controls on individual signatures").

More specific actions taken in regard to the reviewer's comments are written below (in purple) after each comment (in black) and response (in blue).

**Referee #1**

*This paper uses established methods to classify regions with similar physiographic characteristics and with similar flow signatures to determine the best predictive relations at ungauged locations. For this reason, the manuscript reads more as a report rather than a novel contribution to the literature. For example, in lines 13-15, it seems the manuscript goals do not seem to be driven by scientific hypothesis but more by having a large set of data and wanting to develop/explore some relations which may (or may not) be useful at some later point. In this way, I think the motivation for the study seems weak as a scientific contribution. Despite this, I do believe that the novelty of the manuscript is in the application of these methods over such a large spatial domain. As hydrologic modeling efforts expand to cover continental scales, the ability to upscale existing approaches for model calibration across large ungauged regions becomes a limiting factor in these efforts. This point should emphasized more in the manuscript to elevate the impact of the work beyond an application of existing methods to a larger region than had been tested previously.*

The reviewer is completely right in his/her assumption that the scientific contribution of this paper is not in developing any new methods – but applying existing methods to learn more about nature, in this case hydrological controls across Europe. This is emphasized both in the Introduction and in the Discussion and Conclusion of the paper.

Action taken: We have reworked different parts of the introduction to emphasize more this aspect.

*1. The selection of flow signatures needs more details as to how they were selected. Olden and Poff (2003) do not from my remembering of the paper - as the authors indicate in line 16 - provide 9 signatures. Their paper attempts to reduce redundancy in the 200+ statistics that have been used for hydro-ecologic classification but they do not provide a definitive reduced list. More details need to be provided as to why these signatures were selected, particularly because their usefulness in applications is not part of the analysis in the paper. This reads as quite an arbitrary choice.*

Thank you for raising this point, the choice flow signatures was indeed made very carefully in this work. The reviewer is right in claiming that Olden and Poff (2003) do not provide a definitive reduced list of signatures, but they do suggest a way to select such a list when saying "One could reduce the population of indices to a minimum of nine, each of which exhibits the highest absolute loading for the first principal-component axes for each of the nine distinct components of the flow regime (Table III). This ensures that the majority of the variation is accounted for and that different facets of the flow regimes are adequately represented in subsequent analyses. Furthermore, given the particular ecological question being addressed, additional indices within each flow component could be selected (from the remaining significant principal components), which would not result in a substantial increase in redundancy".
According to their suggestion we selected the nine indices with the highest absolute loading for the first principal-component axes for each of the nine distinct components of the flow regime, except for number of zero-flow days describing the component "duration of low events", because this index was to specific to intermittent rivers (a very large majority of the rivers in our domain had a value of 0 for this index). For this component of the flow regime we selected instead the index with the highest absolute loading for the second principal-component axe.

*2. In lines 12-13 (p. 3), the comment is made that this type of analysis has not been applied at the continental scale "including large rivers with human alteration…" Do the catchments examined here have human alteration? This is not noted in the methods? Does this bias your results?*

When visually checking the hydrographs of each flow station, the catchments with obvious and very strong flow regulation where removed. Though, a part of the catchments used in the study still have various forms of human alteration. This has partly been taken into account with some indices like agricultural area, urban area or irrigated area. Unfortunately we haven't been able to find a good indicator of flow regulation available over the whole Europe but this would certainly be of interest if such an index became available. Nevertheless, impact from regulation was clearly identified in the hydrological interpretation of similarities between catchments in specific groups. This is part of the results (Table 3), which is discussed in Section 3.3.

Action taken: Human alterations have been mentioned at the beginning of section 2.1, and altered flow at the end of section 2.1.
Note that some impacts of human alteration are analyzed in the discussion part (p. 20, l. 11-15 of the original manuscript)

*3. In line 18 (p. 8), the statement is made that "identified gauging stations that should be further explored and filtered out…" Was this actually done?*

Thank you for pointing out this imprecision. These stations were filtered out, but no further analysis was done on them yet. The sentence will be modified to make it clearer.

Action taken: the sentence has been changed to "…identified gauging stations that were filtered out for the following analyses."

*4. In Section 2.3, how were variables determined to be significant in the regressions? What diagnostics were used? How many variables were allowed to enter in each equation? It may be useful as an explanatory tool to see which variables are significant but to make predictions (which is the*

*goal of this work), one needs to adhere to good statistical practices. How were these practices followed?*

We agree with the reviewer on the importance of providing this information. The significance testing is described in section 3.1: significance of correlations was tested based on a t distribution with a threshold of 0.05. We agree that this information should be available in section 2.3 as well and will be added l.13. The way the variables (and the number of variables) were selected for each regression is described in section 2.3 p. 8 l. 24-29 (stepwise regression based on the Bayesian Information Criterion). The built regression models were evaluated using statistical measures such as the coefficient of determination. These different steps constitute an established statistical procedure to build and evaluate regression models.

However, we want to point out here that, as stated in the introduction, the main goal doesn't lie in the prediction itself but in gaining better understanding in the hydrological patterns across the European continent. The regression models, like the classifications, are used as a tool to reach this better understanding by exploring the relationships between descriptors and signatures and highlighting the main controls of flow signatures in different types of European catchments.

Action taken:
  - The significance has been defined in section 2.3: "Significance of correlations was tested based on a t distribution with a threshold of 0.05."
  - We have reworked different parts of the introduction and conclusions to better emphasize our actual aim of understanding the hydrological patterns and their main controls.

*5. I found myself questioning the value of Section 3.1. I do not think this offers any additional information beyond what can be determined from the CART and regression analyses. This section also contributes to the manuscript reading more as a report as this section seems to explain what could be characterized as exploratory data analysis that is completed before one settles on an approach and hypothesis to test. I also think that the manuscript is a bit laborious in its reading and removal of this section would help streamline the manuscript.*

Thank you for this suggestion for improving the readability of the paper. This first part of analysis was performed to give a first overview of the links between descriptors and signatures and to closely study the catchment descriptors to decide whether it was reasonable to keep them for further analysis (13 of them were removed). We agree to move most of section 3.1 to the supplementary material and only state the main conclusions of this part of the study in the main text.

Action taken: The section 3.1 has been moved to the supplementary material. This analysis has been mentioned in section 2.3 together with the exclusion of some of the catchment descriptors.

*6. Section 3.2 seems to be missing a reference to how the classification was applied to the data. At the very least, reference Section 2.2 to describe how the classification was completed.*

We agree with this suggestion, the way the classification was applied to the data is indeed described in section 2.2 and a reference to this section will be added in the result section.

Action taken: The first sentence of section 3.1 (former section 3.2) has been modified to refer to section 2.2: "An automatic clustering based on flow signatures was performed first as explained in section 2.2."

*7. I may have missed this but I think it is necessary to develop regression on the flow signatures using the entire dataset to compare to the regression results obtained for the classes. This analysis would determine the objective improvements provided by first classifying the data. If this analysis has been completed, please refer to this in the text when discussing the results.*

The reviewer is right in raising the importance of comparing the regressions obtained for the classes with models calibrated using the entire dataset. This has been done in our study as described in section 2.3 (p. 8 l. 21-24). Both regressions using the entire dataset and regressions obtained for the classes are analyzed in the result section 3.2. As following the reviewer's suggestion, a reference to section 2.3 will be inserted in the result section (p. 12 l. 19).

Action taken: A sentence has been added in the result section (first sentence of new section 3.2) to refer to the method description: "As explained in section 2.3, multiple regression models for signature prediction were developed both using the entire domain and within each group of the three classifications and their results were compared."

*8. There are two papers that I direct the authors to for potential citation. Singh et al. (2014) used CART to classify model parameter behavior across the United States and may be helpful to motivate some of other contexts in which CART has been utilized for model parameterization at ungauged locations. Oudin et al. (2010) ask almost the same question as this paper in how physiographic similarity is related to hydrologic similarity, although they answer this question using actual model results.*

We thank the reviewer for bringing these interesting papers to our attention, they are indeed completely relevant in the context of our work and we will add a reference to them in the revised version.

Action taken: We have included a reference to Oudin et al. (2010) in the introductions section and to Singh et al. (2014) in the methods section.

*Oudin, L., A. Kay, V. Andréassian, and C. Perrin (2010), Are seemingly physically similar catchments truly hydrologically similar? Water Resour. Res., 46, W11558, doi:10.1029/2009WR008887.*

*Singh, R., S.A. Archfield, and T. Wagener, 2014, Identifying dominant controls on hydrologic model parameter transfer from gauged to ungauged basins - a comparative hydrology approach, Journal of Hydrology, doi: 10.1016/j.jhydrol.2014.06.030, 2014.*

**Referee #2 M.C. Westhoff**

*This paper aims to classify a large set of European catchments using a few different regression, and clustering techniques. The results are analyzed by looking at spatial patterns while the main drivers are characterized for each class.*
*Although I personally have no record in catchment classification methods, I judge this paper as potentially publishable. But before that, I think the paper can and have to be improved.*

*The first point I was triggered about was the sentence "So far we have not yet found a widely accepted classification system" (P2, L8), which made me expect that this paper would (or at least aimed) to finalize this issue. However, this is not the case, while I think you can make this attempt by reserving a part of the available dataset for validation. The used dataset is large enough and I think the results would benefit from a "calibration-validation cycle" in which the dataset is split in two randomly chosen sets, of which one is used for calibration and the other for validation. This can be done several times for different randomly chosen subsets. This exercise may tell you more about number of catchments needed in a class and how robust the chosen signatures are.*

We agree that the calibration-validation exercise suggested by the reviewer would be interesting. However, our aim is not to come up with a unified classification system that would have a general application. The main aim of the work is to understand the link between different flow signatures and catchment physiographic attributes and whether these links are different for different groups of catchments that can be defined based on certain characteristics. To this end, we employed different established classification approaches to group catchments and assessed which classification leads to identification of a stronger link.
Based on the reviewer's remark, we feel that our statement about the lack of a widely accepted classification may send a wrong message about the aim of our work. Therefore, we will remove it and try to emphasize that our aim lies in understanding what does control the signatures across a large domain and what we can learn about similarity.

Action taken: we have modified the sentence in the introduction to not give the impression of trying to find a universally accepted classification system.

*A second aspect was that I had problems understanding what was done and in which order. If I am not mistaken, I think you can roughly summarize it by: 1) With a regression analysis catchment descriptors (CD) are correlated with flow signatures (FS). 2) Classes have been derived using 3 different clustering methods: one using CD, one using FS and one using a CART analysis. 3) For each class, correlations between CD and FS are derived and compared with the correlations derived in step 1. If this is indeed the case, I suggest to add e.g. a flow chart and to turn paragraph 2.2 and 2.3 around.*

Thank you for this comment and suggestion that will for sure improve the clarity of the paper. Actually we could write the different steps as follows:
1. correlation analysis giving a first overview of the links between descriptors and signatures and screening of the descriptors (elimination of 13 catchment descriptors without any significant correlation);
2. classification using three different methods;
3. calibration of linear models, on one hand using the whole domain, on the other hand inside each group of the three classifications, and comparison of performance of these different models.

As also raised by Referee #1, the first part about correlation analysis is maybe confusing and a bit redundant so we plan to remove section 3.1, move the graphics to the supplementary material and only state the main conclusions of this part of the study in the main text.
We agree on the suggestion of adding flow chart and will add one.

Action taken:
- A flow chart has been added at the beginning of section 2.
- The section 3.1 has been moved to the supplementary material. This analysis has been mentioned in section 2.3 together with the exclusion of some of the catchment descriptors.

*I very much agree with paragraph 3.4 in which it is suggested that the finding can be used for ungauged basins or to parameterize large scale models. But to really benefit from the results of this paper I would encourage the authors to also publish the regression constants. This would make it possible for others to indeed parameterize large scale models, while other future classification studies can better compare (quantitatively) their results with those of this study.*

We thank the reviewer for his interest on this part of our work; we will publish the regression constants in the supplementary material.

Action: The regression constants have been added in part E of the supplementary material.

*Minor comments: Be consistent in using either the term "Catchment Descriptors" or "physiographic control"*

We will check this again in the revised version.

Action taken: the use of these terms has been homogenized: "catchment descriptors" refers to the 35 different variables used in the study, some of them turning out to be physiographic controls of the hydrological response in some classes.

*P3,L32: Give also the range of the catchment sizes*

That is a good suggestion, we will include this information.

Action taken: the range of area has been included: "…for 35,215 European catchments with a median size (total upstream area of the outlet) of 493 km$^2$, ranging from 1 to 800,000 km$^2$ (Fig. 1)."

*P6,L5: explain what E-HYPE is*

Thank you for pointing out this oversight! This will be added. E-HYPE is a pan-European hydrological model, more information and some model results are available on http://hypeweb.smhi.se/europehype/long-term-means/

Action taken: the sentence has been change to "…2,690 flow gauges across our study domain selected based on agreement between catchment size in metadata and the delineation in the pan-European hydrological model E-HYPE (Donnelly et al, 2012)."

*On P3,L11-12 it is stated that "No study so far, to our knowledge, has applied the results from comparative hydrology at the continental scale, also including large rivers with human alteration and ungauged basins", suggesting that this study will include basin subject to human alteration. Now on P6,L12 it is stated that stations with strong flow regulations were eliminated.*

When visually checking the hydrographs of each flow station, the catchments with obvious and very strong flow regulation where removed. Though, a part of the catchments used in the study still have various forms of human alteration. This has partly been taken into account with some indices like agricultural area, urban area or irrigated area. Unfortunately we haven't been able to find a good indicator of flow regulation available over the whole Europe but this would certainly be of interest if such an index became available. Nevertheless, impact from regulation was clearly identified in the hydrological interpretation of similarities between catchments in specific groups. This is part of the results (Table 3), which is discussed in Section 3.3.

However, your remark, also supported by a comment from Referee #1 let us think that this sentence unnecessarily stresses human alteration when it's not the main object of our study, so we plan to remove the mention to human alteration here.

Actions taken:
- The sentence in the introduction has been changed to "No study so far, to our knowledge, has applied comparative hydrology at the continental scale, therefore including large rivers with human alteration and ungauged basins."
- More details were added about the elimination of stations with strong regulation (end of section 2.1): "This quality assurance mainly eliminated heavily regulated stations, obviously erroneous hydrographs or wrong time steps (e.g. monthly), still keeping stations with moderately altered flow".

*P12,L15: It is unclear to me to which method is referred here. Please clarify*

Thank you for raising this unclarity, we rewrote the sentence as follows: "When looking at the classification based on catchment descriptors, the average of standard deviations of each catchment descriptor within all clusters was estimated to be 0.71, and the average of standard deviations the flow signatures was 0.78. For the CART classification, these numbers are 0.76 for catchment descriptors and 0.67 for flow signatures."

Action taken: as described in the answer.

*P12,L8: You mean actual evaporation, right? Also add this at P13,L15 and potential other locations.*

This is indeed a lack of precision; we mean actual evapotranspiration and will add this precision where relevant.

Action taken: "Actual" has been added where relevant.

*P14,L9: Is it possible to quantify the strong relationship?*

In both works, regression models were built to estimate BFI using geological classification. Both show that the predictive model for baseflow when geological classification was employed were strong, making a conclusion that geology is the determining factor for baseflow estimation. It is therefore difficult to give figures that quantify how strong the relationship is. In the former work (Longobardi and Villani, 2008) they showed the reduction in the prediction error when accurate spatial variability of geology was used in the classification.

**Referee #3**

*(1) The purpose of conducting a correlation analysis seems a bit unclear to me. Firstly, if reducing the number of variables (to be used for classification) was the goal, why is it that only physical descriptors were chosen for culling, and not flow signatures? It could easily be argued that some flow signatures (e.g., HFD, LowFr) which do not have high correlation with most physical descriptors can be removed as well. Secondly, as mentioned in Section 2.2, a PCA is performed anyways to reduce the dimensionality prior to classification. So why prescreen the variables with correlation analysis before applying PCA? Wouldn't PCA alone on the whole dataset (16 flow and 48 physical variables) do the job?*

Thanks for this remark that seems to agree with reflections from reviewers 1 and 2. This first part of analysis was performed to give a first overview of the links between descriptors and signatures and to

closely study the catchment descriptors to decide whether it was reasonable to keep them for further analysis. Flow signatures were selected to describe different components of the hydrological regime (as explained in section 2.1) so all of them were kept along the different steps of the study. However, as the reviewer underlines, some these signatures turned out to have low correlation with most physical descriptors and to be difficult to model. This is an interesting point that we suggest to include in our conclusions for the final manuscript.

The reviewer is completely right when saying that the PCA should be enough to reduce the dimensionality prior to classification; however, the catchment descriptors are used not only for classification but also in the next steps of the study to build linear models and find out which are the main physical controls of flow signatures in different types of catchments. For this analysis it was helpful to have previously removed the less correlated catchment descriptors.

Finally, (also mentioned in our reply to reviewer 2) even though the correlation analysis was a useful introduction for us to start the study, we agree that section 3.1 may be confusing and a bit redundant so we suggest to remove it for the final manuscript, by moving the graphics to the supplementary material and only state the main conclusions of this part of the study in the introduction to Results in the main text.

Action taken:
- More discussion has been added in section 3.4 about which flow signatures were easier or more difficult to model.
- The section 3.1 has been moved to the supplementary material. This analysis has been mentioned in section 2.3 together with the exclusion of some of the catchment descriptors.

*(2) In Section 3.2, the geographical patterns of classification are briefly mentioned for the physical descriptors based classification, and not at all for the flow signatures based one. I think the authors have a huge opportunity here to explain the geographical context of the spatial patterns observed in Fig 3a and b. It is mentioned (Page 11, Lines 13 and 14) that the flow and physical descriptor based classifications lead to different patterns. Why is that? Any speculation on this aspect would be quite helpful here because it directly relates to the main questions asked in this study.*

We thank the reviewer for the comment. We tried to keep this description of the two first classifications short to reduce the overall length of the paper and focus more on the third "combined" classification. We agree that describing and comparing the spatial patterns of the first two classifications would be of interest. However, we were not expecting the two classifications to be similar since the basis of classification for the two are different. Even a classification based only on catchment descriptors could be different if we added or removed some descriptors. The idea we pursued was to try different classifications and get insight into which one gives us more discrimination of the relationships between flow signatures and catchment descriptors and not trying to seek a correspondence between the groups established through the different classification methods. Thus, we suggest to better explain the difficulties for such an analysis in the manuscript but not trying to explain the differences in patterns.

We will, nevertheless, add more discussion on the spatial patterns of the first two classifications when doing the revision.

Action taken:
The introduction has been modified and a sentence has been added to section 3.1 to precise that we are not expecting to find similar patterns with different classification systems:
- In the introduction: "*Furthermore, the two approaches do not necessarily group catchments in the same way since the data sets used for the classification are different. Therefore, one needs to derive functions that link flow characteristics and catchment attributes within each group of catchments classified in either way. Ultimately, we believe that a catchment classification framework has to achieve the advantages both approaches offer to be useful [...].*"
- In section 3.1: "*Correspondence between the two classifications is not expected as the two classifications were performed using different sets of data.*"

*(3) It might be helpful to state the proportion of total area covered by each of the 10 classes obtained through CART (Figure 3c). It is mentioned later in Section 3.4 that the regression models used for predicting flow signatures across Europe perform poorly for classes 3, 6 and 8, and perform best for classes 7, 10 and 11. Knowing the % area of Europe covered by poor and good performing classes would clarify the ability of your classification to predict flow signatures in ungauged catchments. Based on a quick look of Figure 3c, it seems to me that your best performing classes are predominantly clustered around the Alps, and majority of the Europe is covered by the poor performing class 3 (and class 6 covers large areas too). Does this mean that after going through all the efforts of two classifications + CART + regression models, our ability to predict flow signatures at ungauged catchments is only limited to wet, mountainous systems (which we already know from previous studies to be simple and easily predictable hydrological systems)?*

This is indeed a very good comment and we fully agree on the reviewer's suggestion to try to better quantify how much we actually learn from the classification exercise. We suggest including the proportion of total area for the different classes and extending the discussion on this topic.
The class 3, for which we weren't able to distinguish any determinant flow signatures or catchment characteristics, covers 39% of the map area. As pointed out by the reviewer, the classes where the regression models were performing best cover small areas (resp. 2.4, 2.3 and 2% for classes 7, 10 and 11); however, the regression models showed good performances as well for at least some of the flow signatures in classes 1, 4 and 5 which cover a total of 43% of the study area and are not particularly wet or mountainous systems.
Finally, as written in our reply to reviewer #1, we also want to emphasize that the effort of the classifications and regression models wasn't mainly aiming at predicting flow signatures (even though it is an obvious and interesting use) but at gaining better understanding in the hydrological patterns across the European continent, which we were indeed able to do for most of the continent: 61% of the studied area (all classes except class 3). In the next version of the manuscript, we will try to be more precise on what we actually learnt with respect to predictions of flow signatures at ungauged catchments across Europe.

Action taken:
- The percentages of map area covered by each class have been added in Table 3.
- More details (type of catchments) about the "well performing" and "poorly performing" classes have been added in section 3.4.
- The third conclusion point has been developed with more information about the concerned regions.

**Marked-up manuscript**

[revised manuscript text omitted]

---

## Author Response (AR2)

**Understanding Hydrologic Variability across Europe through Catchment Classification – Replies to reviewers, second round**

Anna Kuentz[1], Berit Arheimer[1], Yeshewatesfa Hundecha[1], Thorsten Wagener[2, 3]

[1] Swedish Meteorological and Hydrological Institute, 601 76 Norrköping, Sweden
[2] Department of Civil Engineering, University of Bristol, BS8 1TR, Bristol, UK
[3] Cabot Institute, University of Bristol, UK

*The authors would like to thank all three reviewers for their time reviewing this article and their constructive comments that helped us improving its value.*

**ANONYMOUS REFEREE #1**

In general, I find the maunscript much improved and with minor edits, suitable for publication. I wonder if the publication would be a better fit for the new HESS type "Cutting Edge Case Studies"?

Abstract: Just a minor comment that it's a bit awkward to start the abstract with "we"

*The sentence has been changed from:*
*"We studied physical controls on spatial patterns of pan-European flow signatures by exploring similarities in 16 flow signatures and 35 catchment descriptors for 35,215 catchments and 1,366 river gauges across Europe."*
*to:*
*"Physical controls on spatial patterns of pan-European flow signatures were studied by exploring similarities in 16 flow signatures and 35 catchment descriptors for 35,215 catchments and 1,366 river gauges across Europe."*

- When referencing the class numbers, sometimes 'no.' is used in the text, sometimes "No." and sometimes "class." This needs to be consistent throughout.

*Thanks for this remark, this has been homogenized.*

- In Section 2.3, I wonder why stepwise regression (forward and backward) was not used. This is the much more acceptable practice for developing regression relations and exploring explanatory variables. An even more robust method is found in the R leaps package, which looks at all subsets of the explanatory variables.

*We actually used a stepwise regression (using the R step package) based on the Bayesian Information Criterion (BIC). However, we noticed that this algorithm was sometimes selecting a large number of explanatory variables even though it didn't improve the coefficient of determination .,Therefore, we added a step where we plotted the coefficient of determination  at each step and made our selection based on this plot. We thank you for your recommendation of using the leaps package and will for sure consider this option for our future work.*
*We added "stepwise" in the text.*

- It is important for the authors to note in the text that using CART to narrow the variables used in linear regression may not be useful because CART considers non-linear and monotonic relations between the variables and regression only considers those variables with linear relations. This means

CART could identify variables that are not appropriate to be used in linear regression. This limitation should be noted in the text.

*CART was not used to narrow the variables used in linear regression but to predict, according to catchment descriptors, the classes of the classification based on flow signatures and extend this classification to the whole set of catchments. At the same time, we looked at the variables used in the CART tree to gain better understanding of the main factors driving the separation into hydrologic classes. Adding the information that CART may not be useful to narrow the variables used in linear regressions seem thus confusing to us as it is not what we used CART for.*

- The misclassification rate and number of nodes (up to 20 in some cases) is quite large. How quantitative was the pruning? I missed this information in the text.

*The level of pruning was selected according to several factors. We wanted the three to allow distinguishing all 10 classes (less complex trees didn't make all classes appear) while minimizing the cross-validated error (xerror in the R rpart package). The selected tree had a cross-validated error of 0.69 and a relative error of 0.59.This is written p. 10 l. 3-6 (cross-validation error has been added in the revised version).*

- There should be references included in Table 3 with the descriptions of dominant hydrological processes. Dominant processes themselves are research questions and references should be provided to support these explanations.

*We agree with the reviewer when saying that the dominant processes themselves are research questions. They are indeed one of our main results. However, we don't think the dominant processes described in table 3 need references as they are our own interpretations based on the observation of hydrological regimes, hydrographs and dominant flow signatures and catchment descriptors in each class. These interpretations are then discussed at the end of section 3.3.*

- The conclusions are very nicely done and offer a great summary of the important findings.

*Thank you!*

**REFEREE #2: MARTJIN WESTHOFF**

The authors have improved their manuscript and took away my concerns/critiques about the first version. I only have a few very minor comments prior to publication:

Figure 1: I find it a bit difficult to read this figure. I suggest separating the box 2 and 3 and place arrows in between and potentially do the same with the boxes now separated by the dashed line. I also do not understand why some parts in the analysis box are coloured and others are not. And is 'hydrological interpretation' the same as 'Analysis' or are they two different columns?

*We have now updated the figure according to the suggestions. The coloured boxes in the analysis part correspond to result figures that we have produced and used for trying to answer the questions which are not coloured.*

P11, L20: 'former' and 'latter' is used when there are only two objects. Here you refer to the first out of three objects.
*In this paragraph we are actually talking about only two objects as the paragraph starts with "Only clustering using catchment descriptors or CART can be applied for the whole domain, i.e. in ungauged*

*catchments." We are comparing the classification based on catchment descriptors with the CART classification.*

P11. L 11-13: give some examples of what kind of catchments are in clusters 3 and 11.
*We have added some descriptions of the catchments in each group.*

P19, L2: Write Aridity index instead of AI
*Thank you, we have changed according to your remark.*

P20, L9: You don't necessarily lack understanding of processes; it could also be that you simply don't have the correct data
*The sentence has been changed to:*

*"This highlights the difficulties of understanding process and physical controls to predict low flows with the datasets currently available to us."*

Appendix:
Figure C and D: change 1/3 in the caption to 2/3 and 3/3
Figure M: something is missing in the caption: '(fig. 6 of the m)'
*Thanks, this has been corrected.*

**ANONYMOUS REFEREE #´3**

I think the authors have done a good job in revising the manuscript and have reasonably addressed my comments from the previous peer review round. The manuscript is more compact now and reads much better.

I only found one small typo in the Abstract (Page 1 Line 15), where "the main" is repeated twice. Please correct this error prior to final publication.

*Thank you for your comment, the typo has been corrected.*

[revised manuscript text omitted]